# Intelligent Modeling for Batch Polymerization Reactors with Unknown Inputs

**DOI:** 10.3390/s23136021

**Published:** 2023-06-29

**Authors:** Zhuangyu Liu, Xiaoli Luan

**Affiliations:** Key Laboratory of Advanced Process Control for Light Industry (Ministry of Education), Jiangnan University, Wuxi 214122, China; liuzy95@stu.jiangnan.edu.cn

**Keywords:** intelligent modeling, batch polymerization reactors, state estimation, recursive expectation-maximization algorithm, process fault, sensor data

## Abstract

While system identification methods have developed rapidly, modeling the process of batch polymerization reactors still poses challenges. Therefore, designing an intelligent modeling approach for these reactors is important. This paper focuses on identifying actual models for batch polymerization reactors, proposing a novel recursive approach based on the expectation-maximization algorithm. The proposed method pays special attention to unknown inputs (UIs), which may represent modeling errors or process faults. To estimate the UIs of the model, the recursive expectation-maximization (EM) technique is used. The proposed algorithm consists of two steps: the E-step and the M-step. In the E-step, a Q-function is recursively computed based on the maximum likelihood framework, using the UI estimates from the previous time step. The Kalman filter is utilized to calculate the estimates of the states using the measurements from sensor data. In the M-step, analytical solutions for the UIs are found through local optimization of the recursive Q-function. To demonstrate the effectiveness of the proposed algorithm, a practical application of modeling batch polymerization reactors is presented. The performance of the proposed recursive EM algorithm is compared to that of the augmented state Kalman filter (ASKF) using root mean squared errors (RMSEs). The RMSEs obtained from the proposed method are at least 6.52% lower than those from the ASKF method, indicating superior performance.

## 1. Introduction

Batch reactors are commonly utilized in industries that focus on the production of high-value products with low volume, such as fine chemicals, pharmaceuticals, biochemicals, and food products. The modern trend in today’s competitive economy is to enhance batch operation performance while ensuring consistent and high-quality products. The precision of temperature regulation during the reaction process plays a crucial role in determining the quality and yield of the final products in batch processing [1]. In order to improve the operation of the process and achieve optimal control performance, an accurate model of the batch reactor temperature is required [2,3,4]. The polymerization reaction is the most prevalent batch reaction, and the intermittent reactor, as a critical reactor for polypropylene synthesis, becomes an important element of the control process. However, owing to the peculiarity of chemical production, there are several process control issues in the typical batch polymerization reactor, which manifest as time-delay, nonlinearity, uncertainty, variable coupling, and other issues [5,6]. Compared to traditional model-based control, advanced control has the potential to significantly enhance industrial production. Thus, developing an accurate temperature model for batch reactors and implementing strict temperature control are crucial not only for maintaining process temperature within the desired range, but also for advancing control technology in theory and practice [7,8]. The polymerization process, which is the most critical step in polymer manufacturing, is carried out using a polymerization reactor. As a vital piece of equipment in chemical manufacturing, the reactor has a significant impact on the quality and quantity of chemical products produced. Batch reactors are typically used for the production of chemical products in small batches, a range of types, and extended reaction periods [9,10,11]. An accurate batch reactor model is of great significance to reduce carbon emissions and to meet sustainable development goals (SDG) [12]. However, modeling batch reactors in polymerization reaction engineering is a challenging task due to the complex chemical reaction mechanism and the unpredictable, nonlinear, and time-varying nature of the reaction process [13].

Given these limitations, it is not surprising that significant attention has been given to the development of modeling for batch reactor processes. Mathematical models for chemical processes can be classified into three categories based on the methodologies and ideas used. The first category comprises mathematical models with a strong theoretical foundation, primarily built on relevant physics and chemical principles. In [14], to investigate the thermal behavior of polymerizing particles in various reactor environments, particle stability analysis with population balance modeling is integrated to simulate the dynamics of the entire particle population as they flow. An industrial perspective for the development of polymer reaction engineering models and their application to create new materials, products, and improved or novel processes is proposed in [15]. Membrane reactor modeling, simulation, and operability analysis approaches are introduced in [16]; these approaches are then employed for simulating the polymer membrane reactor unit and performing the operability mapping for identifying the Pareto frontier and redesigning the membrane reactor. The population balance approach is extended to account for heat transfer limitations at the individual particle level. Ref. [17] proposed a steady-state mathematical model for propylene polymerization in slurry and bulk phases. The model incorporates detailed mass, energy, and momentum balances to account for all chemical species and heat transfer mechanisms in the reaction environment. The second type of mathematical model is primarily constructed from data obtained during normal system operation, such as input and output data or experimental outcomes’ data. Then, the data-driven identification methods are employed in batch reactor modeling. For the field of heat transfer in renewable energy systems and prediction of the remaining useful life of lithium-ion batteries, alongside the familiar neural network and fuzzy- and gene-based techniques, emerging ensemble machine learning methods, like Boosted regression techniques, K-means, K-nearest neighbor (KNN), CatBoost, and XGBoost are gaining traction due to their enhanced architectures and ability to handle diverse data types [12,18]. A supervisor design for a pressurized reactor unit in the presence of sensor and actuator faults has been proposed by [19]. A novel approach to modeling using artificial neural networks has been suggested, which enables the quick development of a model from data collected during various batch experiments [20]. Both the aforementioned categories have limitations. Specifically, determining the parameters in mechanism models can be difficult, while data-driven approaches may lack interpretability. The third category of mathematical models combines the benefits of the previous two types of models, creating a hybrid system model based on the combined use of mechanistic and experimental modeling. A key component of this modeling methodology is the utilization of sensor data. After constructing the first principle model, we rely on the information provided by the sensors to accurately estimate the states and identify process faults. A common approach is to treat the process fault as an unknown input (UI). The presence of internal and external factors in complex chemical environments and process industries is inevitable. The model can be adjusted to account for various factors that contribute to its process fault, including system modeling errors and incorrect parameters, by defining them as UIs.

In recent decades, the issue of how to solve UIs has received considerable attention. A common approach to address this issue is to treat UIs either as a persistent bias or a random process with known statistical properties and incorporate them into the system state. This led to the development of the augmented state Kalman filter (ASKF) [21]. An observer was developed to estimate the UIs, and the necessary and sufficient conditions for its existence were presented in [22]. A switched reduced-order state observer design problem with unknown or partially known inputs was formulated to develop a macroscopic traffic stream model [23]. An alternative approach is to model UIs as a parameter conforming to a Markov transition probability, which can be accomplished using a multiple model approach [24]. The machine learning technique has facilitated the identification of UIs and state estimation, and various methods, such as expectation-maximization (EM) and variational Bayesian (VB) techniques, have been utilized for this purpose [25,26]. While the VB approximation is a distribution-estimation method [27], the EM algorithm is focused on point-estimation. Both techniques use iterative optimization to obtain the solutions. The previous category discussed the EM algorithm, which is commonly referred to as batch EM (BEM) and was first introduced by [28]. The fundamental disadvantage of the BEM method is that it is computationally demanding. The recursive implementation of the EM algorithm has lately attracted scholarly interest due to its capacity to minimize computing overhead [29]. The first proposal for a recursive EM (REM) algorithm was made by Titterington [30]. The technique is based on the stochastic gradient approach. To update parameters online, the Fisher information matrix (FIM) of complete data is involved. Cappé derived a stochastic approximation step instead of inverting the FIM to calculate the conditional expectation of likelihood [31,32]. Initially, the REM algorithm was developed to address the parameters of hidden Markov models. Recently, the use of this technique has expanded beyond its original application and it has been employed to address parameter estimation difficulties in the field of system modeling [33].

This article introduces the KF-based REM algorithm, an approach for the real-time state estimation and identification of UIs in batch polymerization reactors. In contrast to conventional methods, the KF-based REM algorithm considers unknown inputs (UIs) as unknown parameters that need to be estimated. At the same time, it treats system states as hidden variables. The algorithm uses sufficient statistics from sensor data to formulate a recursive Q-function for the state-space model in the E-step, enabling simultaneous and recursive estimation of UIs and state variables.Due to its real-time capabilities and potential to reduce computational resources, the KF-based REM algorithm is highly suitable for practical applications, such as state and parameter estimation for large-scale systems. The main contributions of this work can be summarized as follows:Model mismatch is inevitable for batch polymerization reactors because of their complex characteristics and high nonlinearity. The problem of accurate modeling for batch polymerization reactors is addressed in this work.A recursive EM algorithm is derived from the conventional EM algorithm to implement estimation for states and unknown inputs. The recursive EM method has the benefits of real-time and computational efficiency. For the batch polymerization system and some other high-dimensional systems, the recursive EM algorithm is more practical.

## 2. First Principle Modeling of Batch Polymerization Reactor

A small batch process in which several operational stages are executed in a specified process or required sequence is known as an intermittent production process. in general, various reaction phases demand different process performance indexes. The structural features of the intermittent reactor, as the heart of the intermittent manufacturing process, influence the reaction process’ characteristics. The heat transmission time from the heat-conducting oil to the reactor materials is affected by the material and thickness of the reactor wall. The cooling water takes a long time to remove the heat from the materials in the reactor due to the length of the serpentine tube and the thickness of the tube wall, so the intermittent process temperature model will have a long lag time. The intermittent polymerization reaction process, which incorporates energy transfer, material conversion, heat balancing, reaction rate, concentration fluctuations, and other factors, is a complicated manufacturing process. The reactant conversion rate, reactant concentration, reactor temperature, pressure, and other variables, fluctuate during the polymerization reaction, resulting in an intermittent polymerization reaction that is a nonlinear process. Directly deriving an exact mathematical model from the mechanism of intermittent polymerization is nearly impossible. Motivated by the above analysis, an intelligent modeling method is proposed in this paper. To begin, the dynamic process of polymerization reaction and its mechanistic properties are analyzed, then a mathematical model based on reaction kinetics, heat balance, and material balance principles is developed. Then, the modeling errors are handled as UIs. The KF-based REM method is derived to solve the model mismatch problem online.

### 2.1. Process Analysis

Three phases are involved in the polymerization process: a preheating stage, a constant temperature reaction stage, and a cooling stage. Among these, the second step is the most critical and difficult phase to control. The status of the reaction at this stage has a huge effect on the ultimate product quality. Excessive pressure might occur if the reaction temperature is too high. Contrarily, the heat required for the polymerization process may be insufficient at low temperature. If the aforesaid scenario occurs, it will have an impact on not just the product’s quality, but also on the manufacturing process’s safety. Therefore, an accurate model for this stage is very important. This paper mainly focuses on the modeling of the constant temperature reaction stage. A schematic of a 50-L batch polymerization reactor is exhibited in Figure 1; the temperature sensors are also shown in this figure. Consider the dynamic modeling process for polypropylene. The inlet temperature $T_{i}$ and flow $F_{c}$ of cooling water are available as manipulated inputs. The process state variables in the reactor include the material temperature $T_{r}$ and the cooling water temperature $T_{c}$. The equations governing thermodynamic equilibrium in a batch polymerization reactor are presented below
(1)$$\begin{array}{c} \rho VC_{p}\frac{dT_{r}}{dt}=Vk_{0}C_{A}\left(-\Delta H\right)\exp\left(-\frac{E}{RT_{r}}\right)-US\left(T_{r}-T_{c}\right), \end{array}$$
(2)$$\begin{array}{c} \rho_{c}V_{c}C_{pc}\frac{dT_{c}}{dt}=US\left(T_{r}-T_{c}\right)-F_{c}\rho_{c}C_{pc}\left(T_{c}-T_{i}\right). \end{array}$$

Table 1 provides a comprehensive overview of the parameters used in Equations (Equation 1) and (Equation 2), along with their respective explanations and nominal values. A flowchart of the proposed methodology is illustrated in Figure 2.

### 2.2. Problem Formulation

The state space model of batch polymerization reactors is demonstrated by defining the state vector as $x≜{\left[T_{r},T_{c}\right]}^{T}$, which includes the reactor and coolant temperatures, and the manipulated inputs as $u≜{\left[T_{i},F_{c}\right]}^{T}$, which includes the coolant temperature and cooling flow rate. The nonlinear models (Equation 1) and (Equation 2) are linearized around the operational point using Taylor’s expansion to remove second- and higher-order terms. Given that systems often have inherent process noise, it is common to incorporate an additive noise term w(t) into the model formulation:(3)$$\begin{array}{c} \dot{x}\left(t\right)=Ax\left(t\right)+Bu\left(t\right)+Da\left(t\right)+w\left(t\right), \end{array}$$
where
$$\begin{array}{ccc} A & = & \left(\begin{array}{cc} \frac{C_{A}}{\rho C_{p}}\left(-\Delta H\right)k_{0}\exp\left(-\frac{E}{R{\bar{T}}_{r}}\right)\frac{E}{R{\bar{T}}_{r}^{2}}-\frac{US}{V\rho C_{p}} & \frac{US}{V\rho C_{p}}\\ \frac{US}{V_{c}\rho_{c}C_{pc}} & -\left(\frac{\bar{{\bar{F}}_{c}}}{V_{c}}+\frac{US}{V_{c}\rho_{c}C_{pc}}\right. \end{array}\right),\\ B & = & \left(\begin{array}{cc} 0 & 0\\ \frac{\bar{{\bar{F}}_{c}}}{V_{c}} & \frac{{\bar{T}}_{i}-{\bar{T}}_{c}}{V_{c}} \end{array}\right), \end{array}$$

*D* is the unknown inputs matrices with proper dimension, a(t) is the vector of UIs, which represents the process fault, including linearization errors, modeling mistakes, or unexpected circumstances in industrial process, and w(t) denotes the process noise.

And the measurement equation is formulated as
(4)$$\begin{array}{c} y(t)=Cx(t)+v(t), \end{array}$$
where
$$\begin{array}{c} C=\left(\begin{array}{cc} 1 & 0\\ 0 & 1 \end{array}\right), \end{array}$$

With the sampling time $K_{s}$, the continuous-time state dynamic Equation (Equation 3) and the measurement Equation (Equation 4) can be discretized
(5)$$\begin{array}{c} \left\{\begin{array}{c} x_{k}=\Phi x_{k-1}+\Psi u_{k-1}+Ma_{k-1}+w_{k-1}\\ y_{k}=Hx_{k}+v_{k} \end{array}\right. \end{array}$$
where $\Phi \approx I+A\cdot K_{s}$, $\Psi =B\cdot K_{s}$, $M=D\cdot K_{s}$, $k\in N^{+}$ is the sampling instant. The process and measurement noises are represented as $w_{k-1}$ and $v_{k}$, which are both white Gaussian noises with zero mean and known covariance $Q_{k-1}>0$ and $R_{k}>0$, respectively. $a_{k-1}$ is the vector of UIs, which is used to represent the modeling error.

*Notations*: The symbol ′T′ denotes the matrix transpose; ′Tr′ represents the matrix trace, and *E* is the operator of expectation; N(μ,P) represents the multivariate Gaussian distribution with mean μ and covariance *P*. In addition, $\hat{x}$ represents the estimation of the states *x*, $\overset{ˇ}{x}$ denotes the predicted value of *x*, and $\tilde{x}=x-\hat{x}$ denotes the corresponding estimation error. The notation $C\left(x\right)=xx^{T}$ and $D\left(x,P\right)=x^{T}P^{-1}x$ are used to abbreviate the matrix operations involving *x*.

## 3. Recursive Expectation-Maximization Algorithm

The batch expectation-maximization (EM) algorithm is proposed to estimate systems with missing data based on the maximum likelihood principle. The primary objective of the BEM method is to maximize the expected likelihood of complete data concerning latent variables. During each iteration, the algorithm optimizes the likelihood by updating the parameters. In the E-step, the Q-function is typically calculated, representing the expected log-likelihood of complete data:(6)$$\begin{array}{c} Q\left(\Theta ,\Theta^{\prime }\right)=E_{z_{\mathrm{mis}\;}|\left(z_{obs},\Theta^{\prime }\right)}\left\{\log P\left(z_{\mathrm{mis}\;},z_{\mathrm{obs}\;}|\Theta \right)\right\} \end{array}$$
where $z_{obs}$ is the measured dataset, $z_{mis}$ is the incomplete dataset, and the aforementioned sets are subsets of the complete dataset. Θ is the parameter set to be estimated in the iterative step of the current time instant, $\Theta^{\prime }$ is the parameter set already solved in the last time instant, and $\log P\left(z_{\mathrm{mis}},z_{\mathrm{obs}}|\Theta \right)$ is the combination of the measured data and the incomplete data, which is the log-likelihood of the complete data mentioned above. In the M-step, the optimal parameter set Θ is obtained by maximizing the Q-function, which is expressed as:(7)$$\begin{array}{c} \Theta =\arg\max_{\Theta }Q\left(\Theta ,\Theta^{\prime }\right) \end{array}$$

By iteratively executing the preceding two procedures, the Q-function will be maximized locally. Previous studies have shown that the BEM method is convergent.

In addition to the techniques described in Equations (Equation 6) and (Equation 7), a stochastic approximation technique involving recursive computation provides an alternative approach for computing the Q-function of the BEM algorithm. This method replaces the expectation step while maintaining the maximizing step unchanged. To pursue this approach, we present a KF-based REM technique for online state estimation and the identification of UIs in this paper, demonstrating its application in the subsequent derivation.

### 3.1. Recursive Q-Function in the E-Step

By applying the given transformation, the Q-function of the batch EM approach can be expressed as follows:(8)$$\begin{array}{c} \begin{array}{cc} Q\left(\Theta ,\Theta^{\prime }\right)= & E\left\{\log p\left(x_{0}|\Theta_{0}\right)+\sum_{k=1}^{N-1}\log p\left(x_{k}|x_{k-1},\Theta_{k}\right)\right.\\ & \left.+\sum_{k=1}^{N-1}\log p\left(y_{k}|x_{k},\Theta_{k}\right)+\log p\left(x_{N}|x_{N-1},\Theta_{N}\right)+\log p\left(y_{N}|x_{N},\Theta_{N}\right)\right\} \end{array} \end{array}$$

When new data is obtained, Equation (Equation 8) in the REM algorithm should be modified. If the sensor data is collected sequentially over time, the E-step can be performed by incorporating the current sample into the previously estimated parameters at the prior time index. The following derivation is based on this characteristic. Convert Equation (Equation 8) to a quasi-recursive form:(9)$$\begin{array}{c} \begin{array}{cc} Q_{N}\left(\Theta ,\Theta_{N}^{old}\right)= & E\left\{\log p\left(x_{0}|\Theta_{0}\right)+\sum_{k=1}^{N-1}\log p\left(x_{k}|x_{k-1},\Theta_{k}\right)\right.\\ & \left.+\sum_{k=1}^{N-1}\log p\left(y_{k}|x_{k},\Theta_{k}\right)+\log p\left(x_{N}|x_{N-1},\Theta_{N}\right)+\log p\left(y_{N}|x_{N},\Theta_{N}\right)\right\} \end{array} \end{array}$$
the Equation (Equation 9) of $Q_{N}\left(\Theta ,\Theta_{N}^{old}\right)$ is not a conventional recursive Q-function. The subscript *N* here represents the current time instant, and Θ denotes the parameter set to be estimated in the current moment. The variable $\Theta_{N}^{old}$ refers to the outcome solved at time index N−1, which is adopted in computing the posterior expectation at the *N* time instant. The superscript ’old’ denotes that the parameter was estimated at the previous time index *N*. The hidden states are estimated using the parameters obtained recursively through the quasi-recursive Q-function. Accordingly, the quasi-recursive Q-function can be expressed as:(10)$$\begin{array}{c} Q_{N}\left(\Theta ,\Theta_{N}^{\mathrm{old}\;}\right)=Q_{N-1}\left(\Theta ,\Theta_{N-1}^{\mathrm{old}\;}\right)+E\left(\log p\left(x_{N}|x_{N-1},\Theta_{N}\right)+\log p\left(y_{N}|x_{N},\Theta_{N}\right)\right) \end{array}$$

To be more precise, Equation (Equation 10) can be classified as a quasi-recursion because the E-step is updated independently of the time index. However, a true recursive Q-function can be defined at time index *N* as follows:(11)$$\begin{array}{c} {\tilde{Q}}_{N}\left(\Theta ,\Theta_{N}^{\mathrm{old}\;}\right)=\frac{1}{N}Q_{N}\left(\Theta ,\Theta_{N}^{\mathrm{old}\;}\right) \end{array}$$

By substituting (Equation 11) into the aforementioned Equation (Equation 10), a recursive formula for the likelihood of complete data is derived as follows:(12)$$\begin{array}{c} \begin{array}{cc} {\tilde{Q}}_{N}\left(\Theta ,\Theta_{N}^{\mathrm{old}\;}\right)= & \frac{1}{N}Q_{N-1}\left(\Theta ,\Theta_{N-1}^{\mathrm{old}\;}\right)+\frac{1}{N}E\left(\log p\left(x_{N}|x_{N-1},\Theta_{N}\right)\right.\\ & \left.+\log p\left(y_{N}|x_{N},\Theta_{N}\right)\right)\\ = & \left(1-\frac{1}{N}\right){\tilde{Q}}_{N-1}\left(\Theta ,\Theta_{N-1}^{old}\right)+\frac{1}{N}E\left(\log p\left(x_{N}|x_{N-1},\Theta_{N}\right)\right.\\ & \left.+\log p\left(y_{N}|x_{N},\Theta_{N}\right)\right) \end{array} \end{array}$$

To meet the stochastic approximation requirements of $\sum_{N\to \infty }\gamma_{N}=\infty$ and $\sum_{N\to \infty }\gamma_{N}^{2}<\infty$, an artificial step size $\gamma_{N}$ is used in place of the natural step size 1/N in (Equation 12). Consequently, the recursive Q-function of time *N* can be expressed as follows:(13)$$\begin{array}{ccc} {\tilde{Q}}_{N}\left(\Theta ,\Theta_{N}^{\mathrm{old}\;}\right) & = & \left(1-\gamma_{N}\right){\tilde{Q}}_{N-1}\left(\Theta ,\Theta_{N-1}^{\mathrm{old}\;}\right)\\ & & +\gamma_{N}E\left(\log p\left(x_{N}|x_{N-1},\Theta_{N}\right)+\log p\left(y_{N}|x_{N},\Theta_{N}\right)\right) \end{array}$$

To compute the recursion of ${\tilde{Q}}_{N}\left(\Theta ,\Theta_{N}^{\mathrm{old}\;}\right)$ from the initial time index, Equation (Equation 13) can be calculated as follows:(14)$$\begin{array}{c} {\tilde{Q}}_{N}\left(\Theta ,\Theta_{N}^{old}\right)=\prod_{t=2}^{N}\left(1-\gamma_{t}\right)E\left(G_{1}\right)+\sum_{k=2}^{N-1}\left[\prod_{t=k+1}^{N}\left(1-\gamma_{t}\right)\right]\gamma_{k}E\left(G_{k}\right)+\gamma_{N}E\left(G_{N}\right) \end{array}$$
with
(15)$$\begin{array}{c} \begin{array}{cc} E\left(G_{1}\right)= & -\frac{n+(m+n)}{2}\log\left(2\pi \right)-\frac{1}{2}\left(\log\left|Q_{0}\right|+\log\left|R_{0}\right|\right)-\frac{1}{2}\log\left|{\hat{P}}_{0}\right|-\frac{n}{2}\\ & -\frac{1}{2}\mathrm{Tr}\left\{Q_{1}^{-1}\left\{C\left({\hat{x}}_{1}-\Phi {\hat{x}}_{0}-\Psi u_{0}-Ma_{0}\right)+\left({\hat{P}}_{1}-\Phi {\overset{ˇ}{P}}_{1}\right)\right.\right.\\ & \left.\left.-\left({\overset{ˇ}{P}}_{1}\Phi^{T}-\Phi {\hat{P}}_{0}\Phi^{T}\right)\right\}\right\}-\frac{1}{2}\mathrm{Tr}\left\{R_{1}^{-1}\left\{\left(H{\hat{P}}_{1}H^{T}\right)+C\left(y_{1}-H{\hat{x}}_{1}\right)\right\}\right\} \end{array} \end{array}$$
(16)$$\begin{array}{c} \begin{array}{cc} E\left(G_{k}\right)= & -\frac{m+n}{2}\log\left(2\pi \right)-\frac{1}{2}\left(\log\left|Q_{k}\right|+\log\left|R_{k}\right|\right)-\frac{1}{2}\log\left|{\hat{P}}_{k}\right|\\ & -\frac{1}{2}\mathrm{Tr}\left\{Q_{k}^{-1}\left\{C\left({\hat{x}}_{k}-\Phi {\hat{x}}_{k-1}-\Psi u_{k-1}-Ma_{k-1}\right)+\left({\hat{P}}_{k}-\Phi {\overset{ˇ}{P}}_{k}\right)\right.\right.\\ & \left.\left.-\left({\overset{ˇ}{P}}_{k}\Phi^{T}-\Phi {\hat{P}}_{k-1}\Phi^{T}\right)\right\}\right\}-\frac{1}{2}\mathrm{Tr}\left\{R_{k}^{-1}\left\{\left(H{\hat{P}}_{k}H^{T}\right)+C\left(y_{k}-H{\hat{x}}_{k}\right)\right\}\right\} \end{array} \end{array}$$
(17)$$\begin{array}{c} \begin{array}{cc} E\left(G_{N}\right)= & -\frac{m+n}{2}\log\left(2\pi \right)-\frac{1}{2}\left(\log\left|Q_{N}\right|+\log\left|R_{N}\right|\right)-\frac{1}{2}\log\left|{\hat{P}}_{N}\right|\\ & -\frac{1}{2}\mathrm{Tr}\left\{\left\{Q_{N}^{-1}C\left({\hat{x}}_{N}-\Phi {\hat{x}}_{N-1}-\Psi u_{N-1}-Ma_{N-1}\right)+\left({\hat{P}}_{N}-\Phi {\overset{ˇ}{P}}_{N}\right)\right.\right.\\ & \left.\left.-\left({\hat{P}}_{N}\Phi^{T}-\Phi {\overset{ˇ}{P}}_{N-1}\Phi^{T}\right)\right\}\right\}-\frac{1}{2}\mathrm{Tr}\left\{R_{N}^{-1}\left\{\left(H{\hat{P}}_{N}H^{T}\right)+C\left(y_{N}-H{\bar{x}}_{N}\right)\right\}\right\} \end{array} \end{array}$$
where $\prod_{t=k+1}^{N}\left(1-\gamma_{t}\right)$ is the product of the step sizes from k+1 to *N*, and *t* is the corresponding time index. The derivation is detailed in Appendix A.

### 3.2. Kalman Filter

To obtain the posterior values of the states and the covariance in (Equation 14), filter methods are often adopted. The current literature suggests utilizing a fixed-interval smoother for state estimation when applying the EM method. Although the smoother is an effective batch-processing technique, it is not well-suited for online calculations. To overcome this limitation, we utilize a standard KF to estimate the state variables, which can be recursively computed. The KF is specifically designed for linear stochastic systems and offers optimal state estimation, making it a perfect fit for integration with the REM algorithm. The Kalman filter is expressed as follows:

Prediction:(18)$$\begin{array}{ccc} {\hat{x}}_{N} & = & \Phi {\overset{ˇ}{x}}_{N-1}+Ma_{N-1}+\Psi u_{N-1} \end{array}$$(19)$$\begin{array}{ccc} {\overset{ˇ}{P}}_{N} & = & \Phi {\hat{P}}_{N-1}\Phi^{T}+Q_{N-1} \end{array}$$

Update:(20)$$\begin{array}{ccc} K_{N} & = & {\overset{ˇ}{P}}_{N}H^{T}{\left(H{\overset{ˇ}{P}}_{N}H^{T}+R_{N}\right)}^{-1} \end{array}$$(21)$$\begin{array}{ccc} {\hat{x}}_{N} & = & {\overset{ˇ}{x}}_{N}+K_{N}\left(y_{N}-H{\overset{ˇ}{x}}_{N}\right) \end{array}$$(22)$$\begin{array}{ccc} {\hat{P}}_{N} & = & \left(I-K_{N}H\right){\overset{ˇ}{P}}_{N} \end{array}$$

### 3.3. Maximization of the Recursive Q-Function

In order to optimize the recursive Q-function within the KF-based REM algorithm, the algorithm computes the partial derivative with respect to the UIs. The value of the UIs vector $\Theta_{k}$ that maximizes the function is found by setting the partial derivative to zero:(23)$$\begin{array}{c} \frac{\partial {\tilde{Q}}_{N}\left(\Theta ,\Theta_{N}^{\mathrm{old}\;}\right)}{\partial a_{N}}=0 \end{array}$$

From (Equation 23), we can get
(24)$$\begin{array}{c} a_{N}=\left(1-\gamma_{N}\right)a_{N-1}+\gamma_{N}\left(M^{-1}\left({\hat{x}}_{N}-\Phi {\hat{x}}_{N-1}\right)\right) \end{array}$$

**Proof.** 

(25)
$$\begin{array}{c} \frac{\partial {\tilde{Q}}_{N}\left(\Theta ,\Theta_{N}^{old}\right)}{\partial a_{N}}=\frac{\partial }{\partial a_{N}}\left\{\prod_{t=2}^{N}\left(1-\gamma_{t}\right)E\left(G_{1}\right)+\sum_{k=2}^{N-1}\left[\prod_{t=k+1}^{N}\left(1-\gamma_{t}\right)\right]\gamma_{k}E\left(G_{k}\right)+\gamma_{N}E\left(G_{N}\right)\right\}\\ =0 \end{array}$$

Thus, with the derivation of Equation (Equation 24), we conclude the M-step derivation in the KF-based REM algorithm. To outline the proposed approach, the pseudocode is presented in Algorithm 1.    □

**Algorithm 1** Recursive Expectation-Maxization Algorithm**Data:**x^0, P^0, a0, yN, uN, QN, RN, γN**Result:**x^N, aN
  1: **for**
N=1,2,⋯
**do**
  2:   ${\overset{ˇ}{x}}_{N}=A{\hat{x}}_{N-1}+Ma_{N-1}+Bu_{N-1}$
  3:   ${\overset{ˇ}{P}}_{N}=A{\hat{P}}_{N-1}A^{T}+Q_{N-1}$
  4:   $K_{N}={\overset{ˇ}{P}}_{N}H^{T}{\left[H{\overset{ˇ}{P}}_{N}H^{T}+R_{N}\right]}^{-1}$
  5:   ${\hat{x}}_{N}={\overset{ˇ}{x}}_{N}+K_{N}\left(y_{N}-H{\overset{ˇ}{x}}_{N}\right)$
  6:   ${\hat{P}}_{N}=\left(I-K_{N}H\right){\overset{ˇ}{P}}_{N}$
  7:   $a_{N}=\left(1-\gamma_{N}\right)a_{N-1}+\gamma_{N}\left(\mathrm{inv}\left(M\right)\left({\hat{x}}_{N}-A{\hat{x}}_{N-1}-Bu_{N-1}\right)\right)$
  8: **end for**
  9: † **Remarks**: If matrice M is not of full column-rank, it should be computed by using any pseudo-inverse.


## 4. Verification

In this section, we will use Algorithm 1 for the batch polymerization reactors process to illustrate the effectiveness of the suggested technique. To measure the modeling performance, the average root mean square errors (RMSEs) are utilized as the major performance metric. Our goal is to more precisely estimate the states and UIs. Two cases of simulation are adopted to verify the effectiveness of the proposed method.

By inputting the values of the listed parameters in Table 1 into the Equation (Equation 3), the corresponding system matrices can be derived:(26)$$A=\left(\begin{array}{cc} -0.0018 & 0.0028\\ 0.0021 & -0.0086 \end{array}\right),B=\left(\begin{array}{cc} 0 & 0\\ 0.0065 & -0.2083 \end{array}\right),C=\left(\begin{array}{cc} 1 & 0\\ 0 & 1 \end{array}\right),\;$$

In Equations (Equation 5), setting the sampling period $K_{s}=10s$, we can derive
(27)$$\Phi =\left(\begin{array}{cc} 0.9816 & 0.0283\\ 0.0207 & 0.9141 \end{array}\right),\Psi =\left(\begin{array}{cc} 0 & 0\\ 0.0651 & -2.0833 \end{array}\right),H=\left(\begin{array}{cc} 1 & 0\\ 0 & 1 \end{array}\right).$$

In Figure 3 and Figure 4, the practical experimental data denoted as the true value and the state responses of the mechanistic model are compared. The material temperature $T_{r}$ and the cooling water temperature $T_{c}$ in the reactor are the process states, respectively. The initial values of the states are $x_{0}=\left[70,30\right]$ and $x_{0}=\left[80,25\right]$, respectively. The processes last for 3 hrs. As presented in Figure 3 and Figure 4, the state responses of the mechanistic model cannot correspond to the experimental statistics, which is called model-mismatch in the field of system identification. The possible causes of this situation are parameter fluctuations, linearization errors, and even misperception of mechanisms. In modeling, model mismatch is a prevalent issue. The problem now is to estimate the states and the modeling error UIs using the experimental measurements $y_{n}$.

In the final stage of our research, we evaluate the effectiveness and accuracy of our proposed KF-based REM algorithm by comparing it with the existing ASKF method. Figure 5, Figure 6, Figure 7 and Figure 8 show the state responses of the nominal model compensated with UIs, identified using both methods, which demonstrate that both the proposed KF-based REM algorithm and the ASKF algorithm can identify the UIs and estimate states. However, the ASKF method exhibits larger oscillations than the proposed algorithm. To obtain a more comprehensive comparison, we use the average root mean square error (RMSE) as the performance index to evaluate the accuracy of both algorithms. Figure 9 and Figure 10 display the RMSEs of the state estimation using both algorithms. As anticipated, the KF-based REM algorithm provides more accurate results than the ASKF algorithm. Overall, our results suggest that the proposed KF-based REM algorithm is a highly effective and accurate approach for real-time state estimation and identification of UIs in batch polymerization reactors.

## 5. Conclusions

This paper focuses on intelligent modeling for the batch polymerization process. In contrast to traditional modeling methods that are either single-mechanism-based or data-driven, we combine the strengths of both approaches to achieve a more precise solution for batch reactor process modeling. We begin by establishing a first-principle model and then utilize sensor data for process fault detection. In our approach, we model the process faults as unknown inputs (UIs) and employ a modified EM algorithm for online identification of these UIs. The REM algorithm, which we propose, strikes a balance between computational efficiency and identification accuracy. The verification results demonstrate that the REM-KF algorithm accurately compensates for UIs and provides reliable state estimations for batch polymerization processes. This algorithm is proven to be a competitive alternative when the mechanism model falls short of meeting requirements. However, the challenge of selecting the optimal step size $\gamma_{N}$ remains unresolved. Future work may focus on how to obtain the optimal step size γ.

## Figures and Tables

**Figure 1 sensors-23-06021-f001:**
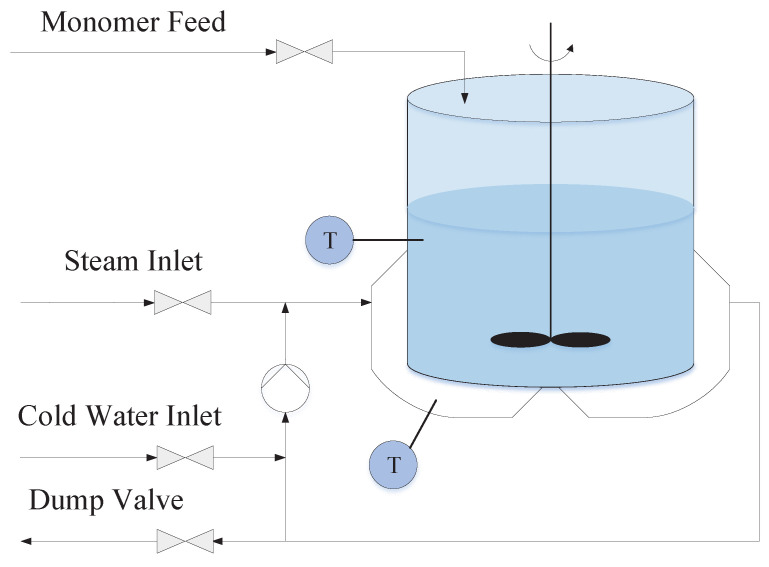
Schematic diagram of the batch reactor.

**Figure 2 sensors-23-06021-f002:**
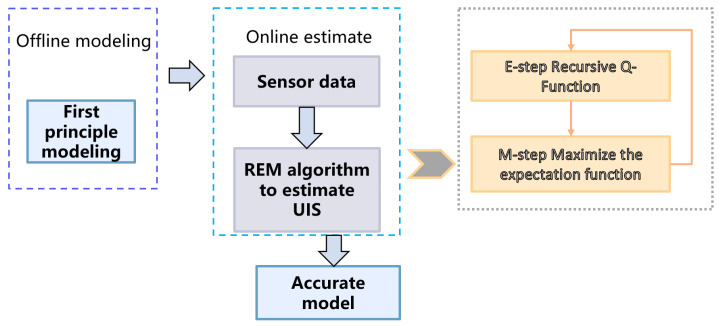
Flowchart of the proposed hybrid modeling method.

**Figure 3 sensors-23-06021-f003:**
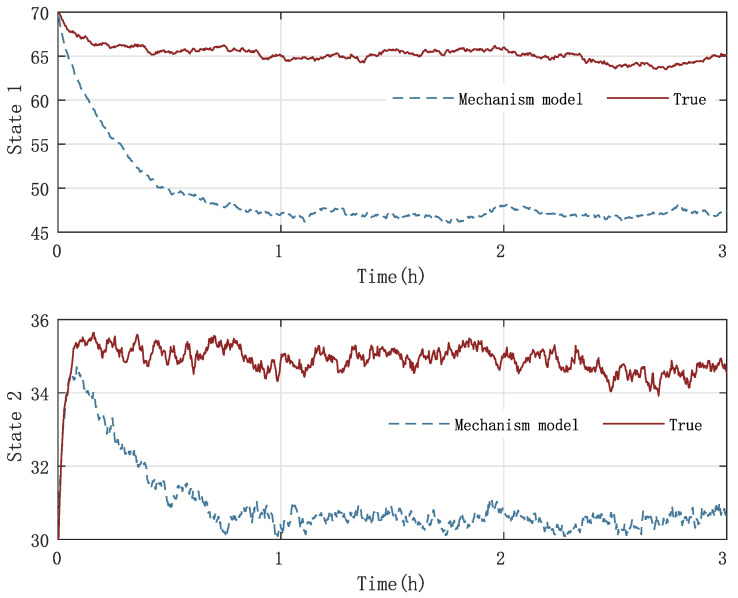
Case 1: Comparison of the experimental data and mechanistic model state responses in polymerization batch reactors: the first state stands for the material temperature, and the second state represents the cooling water temperature.

**Figure 4 sensors-23-06021-f004:**
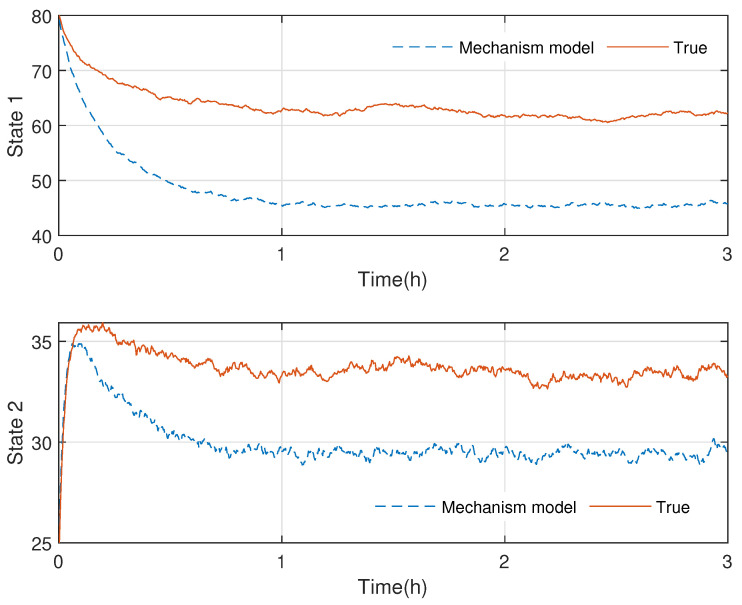
Case 2: Comparison of the experimental data and mechanistic model state responses in polymerization batch reactors: the first state stands for the material temperature, and the second state represents the cooling water temperature.

**Figure 5 sensors-23-06021-f005:**
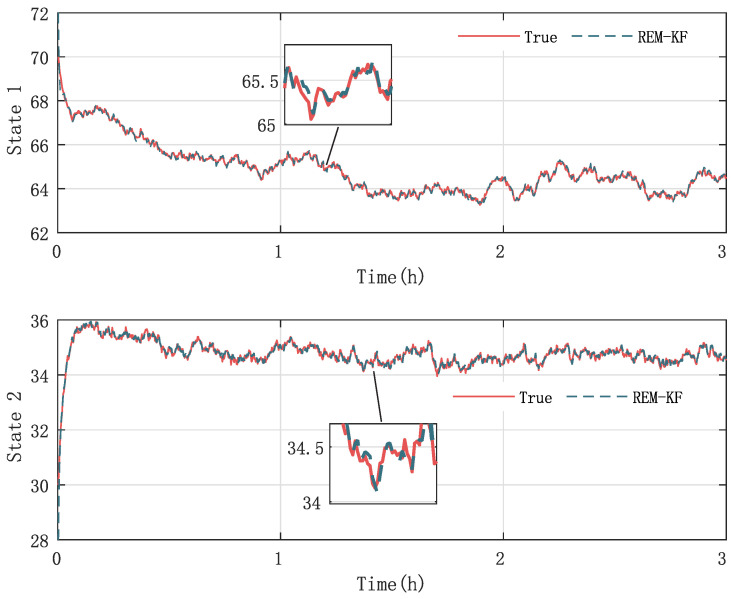
Case 1: State responses of nominal model compensated with UIs identified by the proposed KF-based REM algorithm.

**Figure 6 sensors-23-06021-f006:**
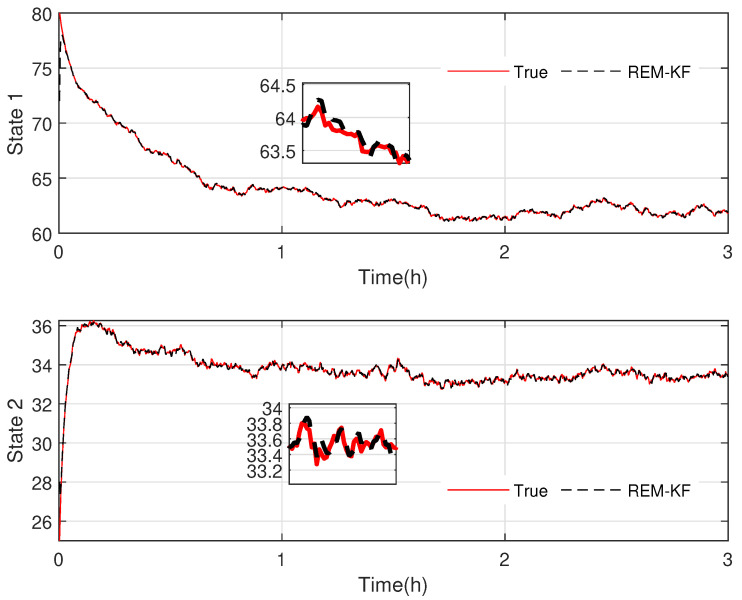
Case 2: State responses of nominal model compensated with UIs identified by the proposed KF-based REM algorithm.

**Figure 7 sensors-23-06021-f007:**
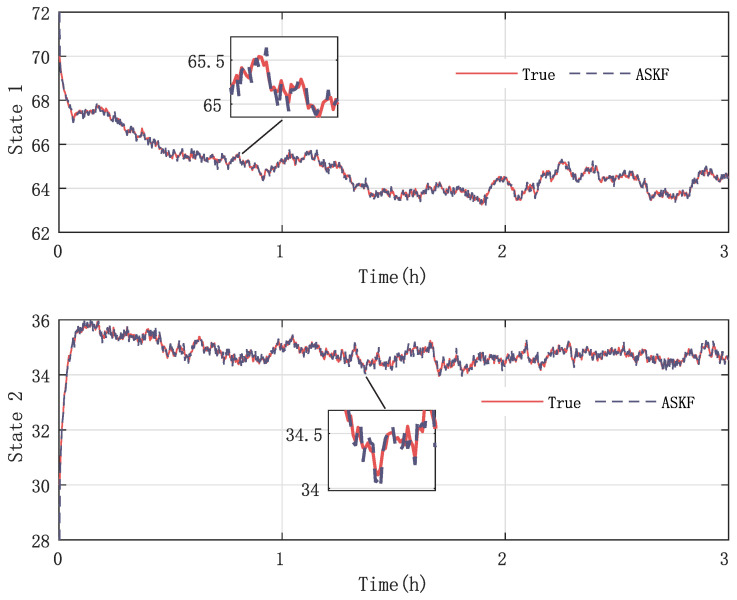
Case 1: State responses of nominal model compensated with UIs identified by the ASKF algorithm.

**Figure 8 sensors-23-06021-f008:**
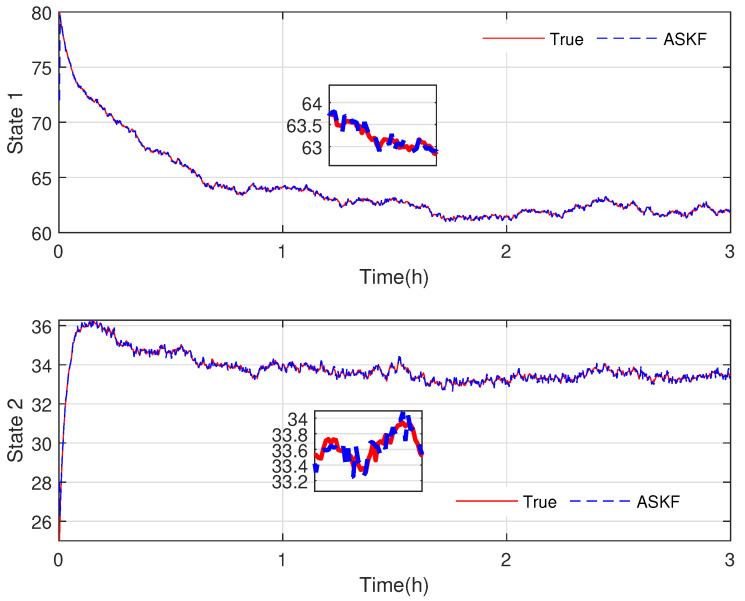
Case 2: State responses of nominal model compensated with UIs identified by the ASKF algorithm.

**Figure 9 sensors-23-06021-f009:**
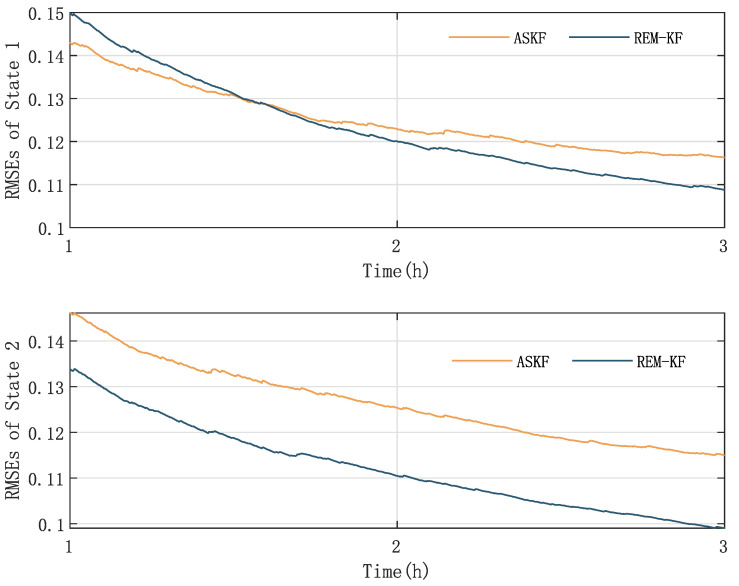
Case 1: The RMSEs of state estimation with ASKF and the proposed algorithm.

**Figure 10 sensors-23-06021-f010:**
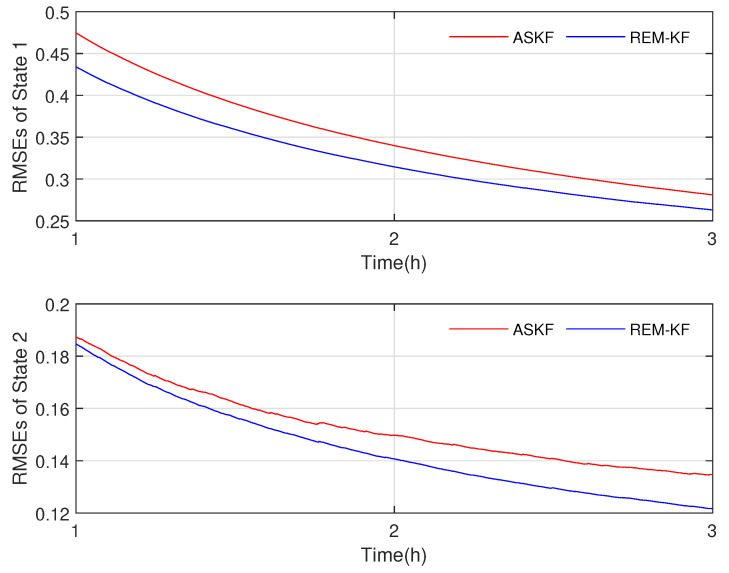
Case 2: The RMSEs of state estimation with ASKF and the proposed algorithm.

**Table 1 sensors-23-06021-t001:** Nominal values of parameters of the batch polymerization reactor.

Parameter	Value	Unit
Heat transfer coefficient *U*	278.6	W/(m^2^·°C)
Heat transfer area *S*	0.75	m^2^
Material volume *V*	35	L
Cooling water volume $V_{c}$	24	L
Material density ρ	$9\times 10^{5}$	g/m^3^
Heat transfer area *S*	0.75	m^2^
Specific heat capacity of material $C_{p}$	2.343	J/(g·°C)
Specific heat capacity of material $C_{pc}$	4.2	J/(g·°C)
Cooling water density $\rho_{c}$	$1\times 10^{6}$	g/m^3^
Material concentration $C_{A}$	11.9	mol/L
Exothermic of reaction (−ΔH)	63.6	KJ/mol
Frequency factor $k_{0}$	$1.25\times 10^{4}$	1/s
Reaction activation energy *E*	$-2.67\times 10^{4}$	J/mol
Gas constant *R*	8.314	J/mol·K

## Data Availability

Not applicable.

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
