# Peer review of "Intelligent Modeling for Batch Polymerization Reactors with Unknown Inputs"

_sensors, 2023, doi:10.3390/s23136021_

Round 1

Reviewer 1 Report

The manuscript discusses an interesting work “Intelligent Modeling for Batch Polymerization Reactors with Unknown Inputs” employing ML for modeling the process of batch polymerization reactors. Following are my suggestions for further improvement:

·       The quantitative outcome of the study should be included in the abstract.

·       The first section of the study should be aligned with global causes like net zero or SDGs.

·       The introduction on intelligent methods may ne improved with the study - Recent advances in machine learning research for nanofluid-based heat transfer in renewable energy system.  

·       The novelty of the study should be highlighted in the last section of introduction. 

·       Kindly avoid long sentences like “In contrast to conventional methods, the KF based REM algorithm considers UIs as unknown parameters that need to be estimated while treating system states as hidden variables.

·       Measuring instruments should also be shown in Figure 1.

·       Used equation in section 2 should be cited.

·       Why the unit of volume is shown as L in table 1.

·       How the uncertainty in modeling measured?

·       Did you investigate hyperparameters of the model?

·       How you prevented model overfitting?

· The machine learning part may be improved with latest work: https://doi.org/10.3390/batteries9010013,%20%20https://doi.org/10.3390/electronics11162534%EF%BB%BF

·       Conclusion must be improved. Also make point wise for easy readability

Overall, it is a well written paper. IT may be accepted after major revision. 

The quality of English is satisfactory.  A thorough check on typos and grammar (specially article) is needed. 

Author Response

We would like to thank the review for the comments with this study. Please see the attachment. 

Reviewer 2 Report

In this paper, a novel recursive approach based on an expectation-maximization algorithm is introduced for identifying actual models for batch polymerization reactors, which is of certain significance. It is recommended that before accepting publication, consider the following recommendations:

1) More information and previous work on
polymerization reactions need to be highlighted in the introduction.
2) Section 2 must be titled as methodology and a flow chart illustrating the proposed method can improve the quality of the paper.
3) Section 3 can be titled an algorithm.

4) What are the Practical Applications of the new approach?
5) A better abbreviation section for significance knowledge is required

6) Provide more data for comparative analysis/verification.

7) Highlight the scientific benefit of this research.

8) The limitations of this research can be mentioned in the conclusions and also the future research.

9) Please correct the minor grammar errors throughout the manuscript

 Please correct the minor grammar errors throughout the manuscript

Author Response

We would like to thank the reviewer for the comments with this study. Please see the attachment.

Round 2

Reviewer 1 Report

The authors have revised the paper well. It may be accepted.